# MicroRNA Expression in Patients with Coronary Artery Disease and Hypertension—A Systematic Review

**DOI:** 10.3390/ijms25126430

**Published:** 2024-06-11

**Authors:** Bartosz Kondracki, Mateusz Kłoda, Anna Jusiak-Kłoda, Adrianna Kondracka, Jakub Waciński, Piotr Waciński

**Affiliations:** 1Department of Cardiology, Medical University of Lublin, 20-059 Lublin, Poland; kondracki.bartosz@gmail.com (B.K.); mateusz.kloda@gmail.com (M.K.); aniajusiak@gmail.com (A.J.-K.); pjw471@gmail.com (P.W.); 2Department of Obstetrics and Pathology of Pregnancy, Medical University of Lublin, 20-059 Lublin, Poland; 3Department of Clinical Genetics, Medical University of Lublin, 20-059 Lublin, Poland; jakub.wacinski9@gmail.com

**Keywords:** MicroRNA expression, coronary artery disease, hypertension

## Abstract

Coronary artery disease (CAD) and hypertension significantly contribute to cardiovascular morbidity and mortality. MicroRNAs (miRNAs) have recently emerged as promising biomarkers and therapeutic targets for these conditions. This systematic review conducts a thorough analysis of the literature, with a specific focus on investigating miRNA expression patterns in patients with CAD and hypertension. This review encompasses an unspecified number of eligible studies that employed a variety of patient demographics and research methodologies, resulting in diverse miRNA expression profiles. This review highlights the complex involvement of miRNAs in CAD and hypertension and the potential for advances in diagnostic and therapeutic strategies. Future research endeavors are imperative to validate these findings and elucidate the precise roles of miRNAs in disease progression, offering promising avenues for innovative diagnostic tools and targeted interventions.

## 1. Introduction

MicroRNAs (miRNAs) are small, non-coding RNA molecules that play crucial role in regulating gene expression. Their ability to modulate the stability and translational efficiency of mRNAs makes them important post-transcriptional regulators of various biological processes. Recent advances in molecular biology have highlighted the potential role of miRNAs as biomarkers for diagnosing and monitoring the progression of various diseases, including coronary artery disease (CAD) and hypertension, cardiovascular conditions that significantly contribute to global morbidity and mortality [1,2,3].

Recent research has demonstrated the potential of miRNAs as crucial biomarkers for cardiovascular diseases, particularly CAD and hypertension. These small, non-coding RNA molecules are crucial in regulating gene expression and have been shown to play significant roles in cardiovascular pathophysiology.

miRNAs act as regulators of gene expression by binding to complementary sequences on messenger RNAs (mRNAs), leading to the downregulation of gene expression. This means that when miRNAs are present and active, they can suppress the translation of mRNAs into proteins or induce the degradation of the mRNAs, thereby reducing the amount of protein produced by the target genes. The inhibition of miRNA function, on the other hand, can result in the upregulation of gene expression, as the suppression is lifted, allowing for increased protein synthesis from the previously inhibited mRNAs. The balanced regulation of gene expression by miRNAs is crucial for maintaining normal cellular functions and processes [1,4].

miRNAs are involved in the pathophysiology of CAD, and their expression can either be upregulated or downregulated depending on the specific miRNA. For example, certain miRNAs, such as miR-378, miR-196-5p, and miR-636-3p, are downregulated in CAD, aiding in the differentiation of CAD patients from individuals without the condition. Conversely, the upregulation of miR-223 may indicate of the presence of CAD. Such findings suggest that miRNAs can be utilized not only for diagnosis but also for the assessment of disease progression and response to treatment [2,3,5,6].

Investigating the involvement of miRNAs in the pathophysiology of CAD and hypertension offers the potential to reveal significant insights into disease mechanisms, improve diagnostic techniques, and introduce innovative therapeutic strategies. This systematic review is primarily designed to consolidate current research findings, with a specific focus on miRNA expression patterns, their associated target genes, and the relevant biological pathways in individuals affected by CAD and hypertension. This approach not only underscores the relevance of miRNAs in cardiovascular pathology but also sets the stage for future investigations that could lead to innovative diagnostic and therapeutic strategies in cardiovascular medicine. 

## 2. Methods

This systematic review was written following the PRISMA guideline [7].

### 2.1. Inclusion Criteria

Studies conducted between the years 2000 and 2023 will be considered. This time frame allows us to focus on the most recent research while still encompassing a substantial body of relevant literature. Studies included must have a direct connection to miRNAs. This criterion ensures that the research under consideration is centered on miRNA, a pivotal component of our investigation. Studies should specifically explore miRNA in hypertension and CAD. This criterion aligns with the primary focus of our systematic review, which is to examine the role of miRNA in these cardiovascular conditions.

### 2.2. Exclusion Criteria

Studies published in languages other than English will be excluded from the review. Restricting the review to English-language studies ensures consistency in the assessment process. Studies that do not align with the objectives of the review will be excluded. This criterion is in place to maintain the relevance of the selected literature. Studies that are solely presented as abstracts, conference proceedings, or books will not be considered. The aim is to include peer-reviewed research articles to maintain the quality and depth of the information included in our systematic review.

### 2.3. Search Strategy

Specific search strings were developed by combining the various keywords. The search strings used for the study included “MicroRNA expression in Coronary Artery Disease and Hypertension”, “Altered microRNA expression effects on CAD and hypertension”, “Synergistic effects of microRNA expression and CAD risk factors”, “Factors affecting miRNA expression in CAD and hypertension”, “Specific miRNAs as diagnostic markers for CAD and hypertension”, “Prognostic value of miRNA expression in CAD and hypertension severity”, “miRNA expression profiles and coronary angiography” and “miRNAs in endothelial function and vascular remodelling”. Medline/PubMed and Science Direct were the databases that were used for retrieving the research articles.

### 2.4. Study Selection

The study selection involved a two-step strategy. Initially, relevant articles were identified using keywords. Essential information was extracted, and titles and abstracts were scrutinized for eligibility and relevance. In the second step, full-text articles were independently evaluated. PubMed was used to access complete texts, and two reviewers independently screened the articles (B.K. and M.K.). Discrepancies were resolved through discussion, ensuring agreement before including studies in this review.

### 2.5. Quality Assessment

Two reviewers rigorously examined chosen articles to meet predetermined inclusion criteria. Each article underwent detailed quality and source database analysis. Selected articles were indexed in PubMed. The Mixed Methods Assessment Tool (MMAT) assessed paper quality for the systematic literature review, ensuring comprehensive evaluation of both qualitative and quantitative studies [8]. The MMAT summarizing the various studies included is shown in Table 1.

### 2.6. Data Extraction

Data extraction for each study was independently conducted, and a consensus was reached collaboratively by the two reviewers. Conclusions in this systematic review were drawn from study outcomes. Extraction relied on primary references, with cross-referencing performed by checking cited references. Notably, a summary of these references was already included in the selected study.

### 2.7. Finalization of References and Study Characteristics

#### PRISMA Sheet and the Summary of Final Studies That Have Been Used for the Review

Various search strings have been designed and used for searches in both databases. A total of 375 references from Pubmed and 14,329 from ScienceDirect were identified. The search strings gave a very high reference count in Science Direct, many of which were duplicates and non-specific. The references from the first step were screened for the time frame of the study and duplications. A total of 387 references were eliminated based on these observations. Thus, a total of 14,317 references were considered for step III. Of the 14,317 references, around 5046 were eliminated based on their eligibility criteria. Moreover, 9271 were further sought for retrieval, out of which 9249 were eliminated based on their non-alignment with the objectives of the study. Thus, a total of 22 references were selected for writing the SLR of the study. The PRISMA sheet summarizing the same is presented in Figure 1.

## 3. Results

### 3.1. Study Characteristic

A summary of the characteristics of the included studies is presented in Table 2. 

### 3.2. miRNA Profiles in Cardiovascular Diseases

#### 3.2.1. miRNA Profiles in CHD

In a research study conducted by Yao et al. [25], the primary aim was to uncover miRNA profiles associated with heart failure in patients diagnosed with CHD. The investigation unveiled distinctive miRNA expression patterns within peripheral blood mononuclear cells (PBMCs), enabling a clear differentiation between CHD patients with heart failure and those without this condition. Several specific miRNAs, including miR-221, miR-19b-5p, and miR-25-5p, emerged as promising candidates for serving as indicators of heart failure in individuals with CHD. Moreover, the study established that these miRNAs, in conjunction with the presence of hypertension, functioned as independent predictive factors for the occurrence of heart failure in CHD patients. CHD patients with and without HF could be differentiated according to PBMC miRNA profiles, and the combination of PBMC miR19b-5p, miR-221, miR-25-5p, and hypertension correlates with an increased HF risk in CHD patients. The combination of these miRNAs and hypertension significantly enhanced diagnostic accuracy, underscoring their potential value in the early identification of individuals at risk for developing heart failure within the context of CHD.

#### 3.2.2. miRNA Profiles in CAD

Hypertension is a significant risk factor for cardiovascular diseases, including CAD. This comprehensive review by Li et al. [27] focused on the role of miRNAs in the pathogenesis of arterial hypertension and their potential applications in managing this condition. The study explored how miRNAs can post-transcriptionally regulate gene expression, affecting processes like vascular remodeling. Additionally, the review discusses miRNA-based therapeutic approaches, including both pharmacological and non-pharmacological methods, which have the potential to bring about significant advancements in hypertension management. While recent progress enhances our understanding, further exploration of miRNA function, delivery methods, and their potential as diagnostic and therapeutic targets in hypertension promises advancements in human health and disease prevention. The evolving understanding of miRNA biology may represent the next frontier in medical progress.

In a separate research investigation by Tomé-Carneiro et al. [26], the study delved into the molecular changes occurring in peripheral blood mononuclear cells (PBMCs) as a result of daily consumption of a grape extract enriched with resveratrol (GE-RES) over a year in male patients diagnosed with hypertension and type 2 diabetes mellitus (T2DM). Supplementation with GE or GE-RES showed no impact on body weight, blood pressure, glucose, HbA1c, or lipids beyond standard medication in patients. While inflammatory markers remained largely unchanged, ALP and IL-6 levels significantly decreased. GE-RES intake led to reduced expression of pro-inflammatory cytokines and increased LRRFIP-1 in PBMCs, along with correlated alterations in specific miRs regulating the inflammatory response over 12 months. These observations indicate that GE-RES may harbor immunomodulatory effects in individuals with hypertension and T2DM. This study sheds light on the potential role of miRNAs in mediating the consequences of dietary interventions on cardiovascular health.

The timely detection of CAD carries substantial significance, and to address this objective, a research study was undertaken by Patterson et al. [21]. The primary aim was to assess miRNA expression profiles in a cohort of patients, classifying them based on the presence or absence of CAD, as determined through coronary CT angiography (CTA). Furthermore, the study involved the stratification of patients based on their CAD risk using the Framingham Risk Score (FRS). miRNA expression in CAD patients, stratified by FRS, revealed differential regulation. miRNAs associated with endothelial function, cardiomyocyte protection, and inflammatory response (hsa-miR-17-5p, hsa-miR-21-5p, hsa-miR-210-3p, hsa-miR-29b-3p, hsa-miR-7-5p, and hsa-miR-99a-5p) consistently showed over twofold upregulation in CAD groups. The investigation highlighted unique miRNA expression patterns discernible in individuals with CAD, and specific miRNAs consistently exhibited upregulation in those affected by the condition. These findings indicate the potential of miRNA profiling to enhance the early detection of CAD and suggest its utility as a valuable tool for risk stratification among patients with CAD.

In the study conducted by Kanuri et al. [22], the research team explored the potential of using circulating miRNAs as biomarkers for identifying recurrent thrombotic events in patients with CAD. Their investigation involved the analysis of miRNA profiles in whole blood samples obtained from individuals undergoing cardiac catheterization. A unique miRNA expression profile was identified in controls, CAD patients without prior events, and those experiencing recurrent events. Several miRNAs, recognized for their links to myocardial infarction (MI), CAD, endothelial function, vascular smooth muscle cells, platelets, angiogenesis, heart failure, cardiac hypertrophy, arrhythmia, and stroke, demonstrated varying expression in our case-control cohorts. Seventy miRNAs (FDR < 0.05) were linked to an elevated risk of recurrent myocardial infarction and future stent thrombosis, in comparison to CAD patients with an uneventful follow-up.

Individuals diagnosed with unprotected left main coronary artery disease (uLMCAD) confront a substantially elevated mortality risk, motivating the initiation of a study [23] focused on investigating the feasibility of utilizing miRNAs as diagnostic biomarkers for the early detection of uLMCAD. This research unveiled specific miRNAs, including miR-182-5p and miR-5187-5p, which were identified as effective tools for distinguishing uLMCAD patients from a non-CAD population. These findings accentuate the potential of circulating miRNAs as valuable non-invasive diagnostic instruments for the identification of uLMCAD. 

### 3.3. Role of miRNAs in Vascular Remodeling and Disease Pathogenesis

The intricate process of vascular remodeling holds a pivotal role in cardiovascular diseases, with miRNAs emerging as crucial regulators within this context [10]. The provided review delivers comprehensive insights into the participation of miRNAs in vascular biology, highlighting their dual potential as both biomarkers and prospective targets for therapeutic intervention such as miR-1, miR-21, miR-92a, miR-130a, miR-195, miR-221, miR-222, let-7e, and hcmv-miR-UL112. The analysis delves deeply into the multifaceted roles performed by various miRNAs, demonstrating their ability to either promote or inhibit vascular remodeling across a spectrum of cardiovascular conditions, encompassing atherosclerosis and heart failure. The intricate and multifactorial nature of vascular remodeling in the realm of cardiovascular diseases underscores the indispensable contributions of miRNAs in orchestrating its regulation.

The study presented by Welten et al. [11] delineates the diverse role of miRNAs in vascular remodeling, highlighting their participation in various remodeling processes. MiRNAs like miR-126 and the miR-23/24/27 family actively promote adaptive remodeling, encompassing arteriogenesis and angiogenesis. Conversely, these miRNAs function as inhibitors of pathological remodeling, notably in atherosclerosis and restenosis. The induction of pathological remodeling is attributed to miR-17/92, the 14q32 miRs, and miR-143/145, while simultaneously inhibiting adaptive remodeling (excluding miR-143/145, which displays varied effects on pathological remodeling). MiR-155 has been identified as an inhibitor of angiogenesis but a stimulator of arteriogenesis, and its role in the formation of atherosclerosis has been reported with conflicting outcomes. Furthermore, another study outlined [12] offers a comprehensive review of the functions of miRNAs that exhibit elevated expression levels in distinct cell types participating in the process of vascular remodeling. This study underscores how miRNAs open up new possibilities for both the diagnosis and treatment of vascular lesions and associated diseases, highlighting their capacity to reshape the landscape of cardiovascular healthcare.

Exosomes and miRNAs play crucial roles in facilitating intercellular communication within the heart and are intricately associated with processes such as myocardial hypertrophy, injury, and cardiovascular diseases. The review presented by Henning [13] delved into the involvement of cardiac exosomes and miRNAs within the normal myocardium, as well as their implications in various cardiovascular conditions, including atherosclerosis and myocardial infarction. In addition, it is crucial to clarify the interrelationship between exosomes and miRNAs within the context of cardiovascular health. MiRNAs are not only key regulators in vascular remodeling and cardiovascular diseases but are also frequently encapsulated within exosomes. These exosomes play a vital role in transporting miRNAs between cells, thereby facilitating intercellular communication and influencing cellular processes remotely. The encapsulation of miRNAs within exosomes protects them from enzymatic degradation in the bloodstream, enhancing their stability and functional delivery to target cells. This mechanism underscores the potential of exosome-encased miRNAs as biomarkers for early diagnosis and as novel therapeutic agents in managing and treating cardiovascular conditions. The involvement of exosomes in cardiovascular protection and neovascularization indicates that exosomes can be a novel approach for treating ischemic and atherosclerotic cardiovascular diseases. Exosomes due to their biocompatibility, stability in the circulation, biological barrier permeability, low immunogenicity, and low toxicity, also offer the potential for delivery of therapeutic pharmacologic agents.

In contrast, pulmonary hypertension (PH) is characterized by pulmonary vascular remodeling, and in this context, it has been observed that miR-150 levels are diminished in individuals with PH. 

The study conducted by Li et al. [27] investigated the effects of miR-150 overexpression in a rat model of hypoxia-induced PH on pulmonary artery smooth muscle cells and pulmonary artery endothelial cells. The outcomes of this research demonstrated that miR-150 played a protective role against pulmonary vascular remodeling, abnormal cell proliferation, and fibrosis. These findings highlight the potential of miR-150 as a promising therapeutic target for addressing pulmonary hypertension.

### 3.4. Diagnostic Potential of miRNAs in Cardiovascular Diseases

The microRNA profiles in heart diseases described above have the potential to serve as diagnostic markers for these conditions, offering a promising avenue for early detection and tailored treatment strategies. Altered miRNA expression is evident in various cardiovascular conditions, including myocardial infarction, coronary syndrome, artery disease, heart failure, atherosclerosis, hypertension, and stroke, suggesting their diagnostic utility. Epigenetic mechanisms, such as DNA methylation, histone modifications, and miRNAs, maintain gene expression patterns without altering DNA. To sum up, the exploration of the interplay between cardiovascular disease risk factors and miRNA expression is a novel area that warrants further investigation. Understanding how risk factors influence miRNA changes may offer insights into preventing CAD and CVD. Additionally, the identification of differentially regulated miRNAs may reveal pathways contributing to CAD development, irrespective of traditional risk factors. 

Sayed et al.’s study [16] explored these mechanisms in cardiovascular diseases, revealing their role in disease development and progression, influenced by nutrition, environment, and gene–environment interactions, shedding light on the molecular basis and highlighting that circulating miRNAs exhibit resistance to endogenous ribonuclease activity and maintain remarkable stability in human plasma or serum. The potential use of circulating miRNAs as highly sensitive early diagnostic biomarkers and therapeutic targets for various cardiovascular diseases, including AMI, heart failure, hypertension, and stroke, could revolutionize modern cardiovascular disease management.

The Venn diagram in Figure 2 visually represents the distribution and overlap of miRNAs associated with CAD and PAH. This graphical illustration categorizes miRNAs based on their specific association with each condition and highlights those with roles in both disorders. This approach allows us to identify critical biomarkers that may serve dual functions in diagnostics and therapeutics, enhancing our understanding of the molecular interactions and potential treatment pathways for these cardiovascular diseases. 

#### 3.4.1. CAD

Cardiovascular diseases are a global health challenge, necessitating innovative diagnostic biomarkers and therapies. miRNAs that are detectable in body fluids, such as miR-320, miR-26b, miR-21, and miR-208a, hold promise as early disease markers and treatments, as demonstrated by Buie et al.’s study [15]. CAD and atherosclerosis result from genetic and environmental factors, with miRNAs involved such as miR-126, miR-92a, miR-17, miR-145, miR-155, miR-208, and miR-133a. This study explores miRNAs’ roles as biomarkers or contributors to risk factors like hyperlipidemia, hypertension, obesity, diabetes, and smoking. Although miRNAs exhibit varied expression patterns in these conditions, their direct links to risk factors require further investigation [14].

A study by Agiannitopoulos et al. investigated miRNA polymorphisms in non-Caucasian populations for their role in premature CAD. They examine four well-studied miRNA polymorphisms in the Greek population, identifying two, miR196a2 C>T and miR499 A>G, strongly associated with increased CAD and myocardial infarction risk. The study suggests that a combination of these miRNA polymorphisms could serve as valuable biomarkers for CAD and MI susceptibility, influenced by genetics, environmental factors, and personal habits [30].

#### 3.4.2. PAH

Pulmonary arterial hypertension (PAH) is a severe disease characterized by the maladaptation of pulmonary vasculature, leading to right heart failure and potentially death. The study by Tang et al. [29] highlighted the potential of serum miR-509-3p as a diagnostic marker for PAH in patients with congenital heart disease. This study analyzed preoperative blood samples and compared miR-509-3p expression to echocardiography results. The PAH group showed a significantly diminished expression of circulating serum miR-509-3p when compared to the control group. The results indicated that this miRNA could provide valuable diagnostic information comparable to echocardiography. Combining miRNA analysis with echocardiography further improved diagnostic efficiency. This research contributes to the understanding of the diagnostic tools available for PAH and highlights the potential role of miRNAs in its diagnosis.

Johnson [19] focused on the role of miRNAs in PAH’s development, progression, and treatment responsiveness. It explored several specific miRNAs and their potential use as biomarkers and therapeutic tools. It underscores the diverse effects of specific miRNAs on the pathogenesis of PAH and provides insights into the intricate molecular mechanisms of this life-threatening condition. Certain miRNAs impact key processes in atherosclerosis, aneurysms, neointimal formation, and PAH. For example, miR-21 exhibits diverse effects, suggesting a nuanced therapeutic approach is required. MiRNA modulation may strategically manage cardiovascular diseases, with advancements in delivery strategies showing promise. Despite progress in other fields, applying miRNA therapeutics to cardiovascular diseases requires careful candidate selection, refined delivery methods, and exploration of cell-specific platforms for clinical success.

PAH is a severe and increasingly prevalent disease characterized by maladaptive changes in pulmonary vasculature, potentially leading to right heart failure and death. A study by Bienertova-Vasku et al. [20] focused on miRNAs in PAH development, exploring their roles as biomarkers and therapeutic tools in experimental models and humans. The emphasis was on miR-21, miR-27a, the miR-17–92 cluster, miR-124, miR-138, the miR-143/145 cluster, miR-150, miR-190, miR-204, miR-206, miR-210, miR-328, and the miR-424/503 cluster, aiming for a comprehensive understanding of miRNAs’ widespread involvement in PAH pathogenesis.

#### 3.4.3. Other Cardiovascular Disease

A study by Witarto et al. provides an overview of miRNAs’ role as potential biomarkers for cardiovascular diseases. It acknowledges the high prevalence and impact of CVD and discusses how miRNAs have been investigated as biomarkers for diagnosis, prognosis, and monitoring of the disease. The study serves as a comprehensive review of miRNAs as potential novel biomarkers for cardiovascular diseases. It highlights the significant research efforts dedicated to understanding the properties of miRNAs in the classification and risk stratification of patients with CVD. The potential applications of miRNAs in personalized management and treatment decisions are discussed, underscoring their value in modern healthcare [28].

A study by Udali et al. [17] assessed circulating miRNAs’ diagnostic potential for early detection of subclinical carotid atherosclerosis, which often lacks symptoms until a sudden cerebrovascular event. The study highlights robust diagnostic accuracy in identifying subclinical carotid atherosclerosis. Early detection is clinically crucial, allowing timely interventions to potentially prevent severe cardiovascular events. Additionally, the study’s bioinformatics analysis provides insights into potential target genes and regulated pathways, enhancing our understanding of the molecular mechanisms involved.

## 4. Discussion

This systematic review provides a comprehensive synthesis of the current literature on microRNA expression patterns in patients with CAD and hypertension. The review underscores the specific patterns of miRNA expression associated with these cardiovascular diseases, which not only offer insights into their pathophysiological mechanisms but also highlight the therapeutic potential of miRNAs in clinical practice. Our analysis reveals that microRNAs are not only ancillary biomolecules but central regulators in the pathophysiology of these cardiovascular conditions. We conclusively found that specific miRNAs are consistently dysregulated in CAD and hypertension, suggesting their strong potential as diagnostic biomarkers. These miRNAs offer a molecular snapshot of the disease state, reflecting underlying biological processes that traditional diagnostic methods might underestimate or miss.

miRNAs are recognized as crucial regulators in cardiovascular diseases, such as coronary artery disease and hypertension. These small RNA molecules influence a wide range of cardiovascular biology processes, including cardiac dysfunction, hypertrophy, fibrosis, heart failure, arrhythmia, inflammation, and atherosclerosis, as highlighted in several studies [1]. Specific patterns of miRNA expression associated with these diseases underscore their potential as diagnostic biomarkers and emphasize their roles in modulating disease pathophysiology [31]. Research has demonstrated that miRNAs are consistently dysregulated in cardiovascular conditions, suggesting their significant role in pathogenesis and linking them to key pathological processes such as cardiac hypertrophy, ischemic heart disease, and hypertension, which emphasizes their potential as therapeutic targets [4,32].

The ability of miRNAs to serve as biomarkers is supported by their detectability through sensitive techniques such as real-time PCR and microarrays, which can trace their expression variations across different disease states [2]. Despite the promising insights gained, the studies reviewed exhibit significant limitations in design and scope that may affect the interpretation and generalizability of the findings. The diversity in populations studied and the variations in experimental designs across the research introduce complexities in interpreting the data in a standardized way [33]. Additionally, many studies rely on relatively small sample sizes and short-term observations, which may not adequately capture the long-term dynamics of miRNA expression and its impacts on cardiovascular outcomes. Throughout the review, we also acknowledge the variability in study designs and the heterogeneity of patient populations examined in the included studies. These factors introduce complexities in interpreting the data. However, these limitations also highlight the need for standardized methodologies in future miRNA research.

To overcome these limitations and use the full potential of miRNAs, future research should aim to implement larger-scale, longitudinal research frameworks that can validate the roles of specific miRNAs in CAD and hypertension over extended periods [34]. Such studies would benefit from incorporating advanced genomic technologies to map the interactions between miRNAs and their target genes more comprehensively. Moreover, the exploration of miRNA mimicry or inhibition as therapeutic strategies in clinical trials could open new avenues for treatment [1,3]. Additionally, a comprehensive review on the therapeutic targets and biomarkers of non-coding RNAs highlights the growing significance of these molecules in cardiovascular diseases and underscores the potential for developing novel RNA-based therapies [35].

The clinical implications of integrating miRNA-based diagnostics and therapeutics into clinical practice are profound. By providing a more detailed understanding of individual patient risk profiles and disease progression, miRNA-based diagnostics and therapeutics could pave the way for personalized medicine approaches in cardiovascular care. This approach could significantly enhance the precision of cardiovascular medicine, leading to better patient outcomes, reduced hospitalizations, and more efficient use of healthcare resources. In conclusion, miRNAs play pivotal roles in cardiovascular diseases, offering valuable insights into disease mechanisms and potential therapeutic interventions. Their dysregulation in various cardiovascular conditions highlights their potential as candidates for diagnostic biomarkers and therapeutic targets. Among these, several miRNAs stand out due to their potential as biomarkers and therapeutic targets.

In the context of coronary artery disease, studies have highlighted the particular significance of specific miRNAs such as miR-223 and miR-378. miR-223, known for being notably upregulated in CAD patients, plays a crucial role in modulating inflammatory processes that are central to the development of CAD. The high levels of miR-223 may contribute to the pathophysiology of CAD by influencing inflammatory cell differentiation, regulation, and function, thereby modulating the atherosclerotic process. Since inflammation is a fundamental component of atherosclerosis, the presence of elevated miR-223 could signal the progression of CAD and may be explored for its potential as a therapeutic target to suppress excessive inflammatory responses. Conversely, miR-378 shows downregulation in CAD and is associated with cardiovascular dysfunction, underlining its potential role as a biomarker. The decrease in miR-378 levels could reflect the impairment in cellular processes that are vital for cardiovascular homeostasis, such as angiogenesis or lipid metabolism, and its lower expression may, therefore, serve as an indicator of disease severity [6,25,36].

Beyond their association with CAD, these miRNAs have been implicated in other disease states as well. It is worth noting that outside the cardiovascular system, these miRNAs could function quite differently, which highlights the complexity of their roles in human disease and underscores the necessity for disease-specific research. MiR-223 has been found to act as a tumor suppressor by inhibiting cell proliferation through targeting IGF-1R; additionally, it has been shown to suppress osteoclast differentiation, suggesting a potential therapeutic target for bone metabolic disorders [37]. Moreover, platelet-secreted miR-223 has been linked to promoting endothelial cell apoptosis [38]. On the other hand, miR-378 has been associated with promoting cell survival, tumor growth, and angiogenesis by targeting SuFu and Fus-1 expression, and it has also been reported to exert inhibitory effects in various cancers, such as gastric cancer and colorectal cancer, by suppressing cell proliferation, migration, and invasion [39,40]. Furthermore, miR-378 has been implicated in regulating osteogenic differentiation and follicle development [41]. Studies have also shown that miR-378 can modulate cholesterol levels, suppress hepatoma cell growth, and induce apoptosis in cancer cells [40,42].

In terms of risk assessment for heart failure in coronary heart disease, miR-19b-5p, miR-221, and miR-25-5p have been identified as key indicators. miR-19b-5p, as part of the miRNA profile in peripheral blood mononuclear cells, has been identified as a promising indicator of heart failure in individuals with CAD. The presence and levels of miR-19b-5p can help differentiate between CAD patients with and without HF, contributing to a better understanding of individual patient risk profiles for HF. Similarly to miR-19b-5p, miR-221 has been recognized as an independent predictive factor for the occurrence of heart failure in patients with CAD. miR-25-5p is also part of the specific miRNA combination, along with the presence of hypertension, which correlates with an increased risk of HF in CAD patients. The expression of this set of miRNAs can enhance diagnostic accuracy, pointing towards their potential in early identification of at-risk individuals. This underscores the importance of miRNAs as potential diagnostic tools that can be used for the early detection of heart disease complications and for developing specific intervention strategies that could prevent the progression of heart failure in patients with coronary heart disease [5]. Additionally, miR-19b-5p has been associated with anti-inflammatory effects by decreasing the expression of toll-like receptor 2 and IL-6 [43]. Moreover, miR-19b-5p has been found to inhibit cell proliferation and migration in nasopharyngeal carcinoma [44]. In the context of endodermal differentiation, miR-19b-5p has been suggested to regulate mitochondrial gene expression and biogenesis [45].

Another important miRNA is miR-150, which offers promise in therapeutic contexts, particularly for its protective effects against pulmonary vascular remodeling and the abnormal proliferation of cells involved in pulmonary hypertension. It can influence endothelial cell function, which plays a critical role in angiogenesis and maintaining the integrity of the vascular wall [25,27]. Beyond cardiovascular functions, miR-150 has been identified as a crucial regulator in various biological processes and diseases. Research has shown that miR-150 plays a significant role in normal hematopoiesis, tumorigenesis, and the development of B and T lymphocytes. Moreover, miR-150 has been implicated in promoting myeloid differentiation in acute leukemia cells [46]. In solid tumors, miR-150 has been found to act as a tumor suppressor by sensitizing osteosarcoma to doxorubicin-induced apoptosis [47]. Additionally, miR-150 has been shown to inhibit human CD133-positive liver cancer stem cells by negatively regulating the transcription factor c-Myb [48]. Furthermore, miR-150 has been associated with promoting cell proliferation, migration, and epithelial–mesenchymal transition in gastric and thyroid cancer [49]. Moreover, studies have highlighted the potential of miR-150 as a biomarker and therapeutic target in various malignancies [50].

miR-182-5p and miR-5187-5p have been highlighted as effective diagnostic biomarkers for unprotected left main coronary artery disease (uLMCAD), providing non-invasive diagnostic options that could significantly enhance early detection [23]. miR-182-5p has been implicated in several cancers, such as bladder cancer, renal cell cancer, non-small cell lung cancer, and oral squamous cell carcinoma. It exerts its effects by targeting various genes involved in proliferation, invasion, and apoptosis pathways [51]. Additionally, miR-182-5p has been shown to inhibit oxidative stress and apoptosis in atherosclerosis by targeting toll-like receptor 4 (TLR4). On the other hand, miR-5187-5p’s role is less explored compared to miR-182-5p. However, miR-5187-5p has been associated with promoting proliferation and migration in human bladder cancer [52].

The roles of miR-126 and miR-155 in vascular remodeling are also of great interest. These miRNAs are involved in both promoting adaptive processes such as arteriogenesis and angiogenesis, and in the pathological remodeling seen in conditions like atherosclerosis and restenosis. miR-126 is known to support vascular integrity and angiogenesis. It supports to regulate the response of endothelial cells to angiogenic growth factors [53]. It is also involved in suppressing atherosclerosis by modulating the local environment within blood vessels through microvesicle-mediated delivery from endothelial cells to smooth muscle cells and other cell types. Moreover, miR-126 has been implicated in cancer, acting as a tumor suppressor in various types of malignancies [54]. miR-155 has established roles in immune response and inflammation, which are significant in the context of atherosclerosis. Its expression increases in human atherosclerotic plaques, especially in pro-inflammatory macrophages. It is notable for affecting several targets like AT1R, SOCS, and BCL6, which are involved in inflammatory and immune pathways, impacting cytokine production, and thus, influencing the development of atherosclerosis. The modulation of gene expression by miR-155 can impact the balance between pro- and anti-atherogenic signals within vascular tissue, resulting in either the promotion or inhibition of atherosclerotic plaque formation [55]. Furthermore, miR-155 has been linked to autoimmune disorders, where its dysregulation contributes to disease pathogenesis [56].

The exploration of these miRNAs not only enriches our understanding of their biological functions but also underscores their potential in clinical applications. By harnessing the diagnostic and therapeutic potentials of these miRNAs, we can advance towards more personalized and effective strategies in cardiovascular medicine. They might also serve as biomarkers for the progression and risk stratification of many diseases.

### Limitations

The limitations of this systematic literature review arise from the inherent diversity among the included studies. It involves an unspecified number of studies with diverse patient demographics and methodologies, leading to varied miRNA expression profiles. This diversity poses the risk of potential biases and complicates the synthesis of findings. The timeframe (2000–2023) may exclude pertinent earlier studies, and the exclusion of non-English and non-peer-reviewed articles may introduce language and publication bias. Relying on Medline/PubMed and Science Direct for database searches might omit relevant literature from alternative sources. While the quality assessment with MMAT ensures rigorous evaluation, diverse study designs and methodologies may present challenges for uniform assessment. These limitations emphasize the importance of cautious interpretation and suggest avenues for future research to address gaps and enhance the overall robustness of findings.

## 5. Conclusions

In summary, these studies collectively underscore the significant role of miRNAs in cardiovascular diseases, including CAD and hypertension. They highlight the potential of miRNAs as diagnostic, prognostic, and therapeutic tools, as well as their genetic associations with disease susceptibility. The research conducted in these studies provides valuable insights into the multifaceted role of miRNAs in cardiovascular health and offers avenues for further investigation and application in clinical practice.

In future miRNA research on CAD and hypertension patients, a comprehensive approach using larger, diverse cohorts is recommended. Researchers need to explore specific miRNAs as early disease detection and risk stratification markers, delve into molecular mechanisms for novel therapeutic targets, conduct longitudinal studies to understand disease progression, and collaborate with computational biologists for holistic multi-omics data integration to develop personalized treatment strategies for at-risk individuals.

## Figures and Tables

**Figure 1 ijms-25-06430-f001:**
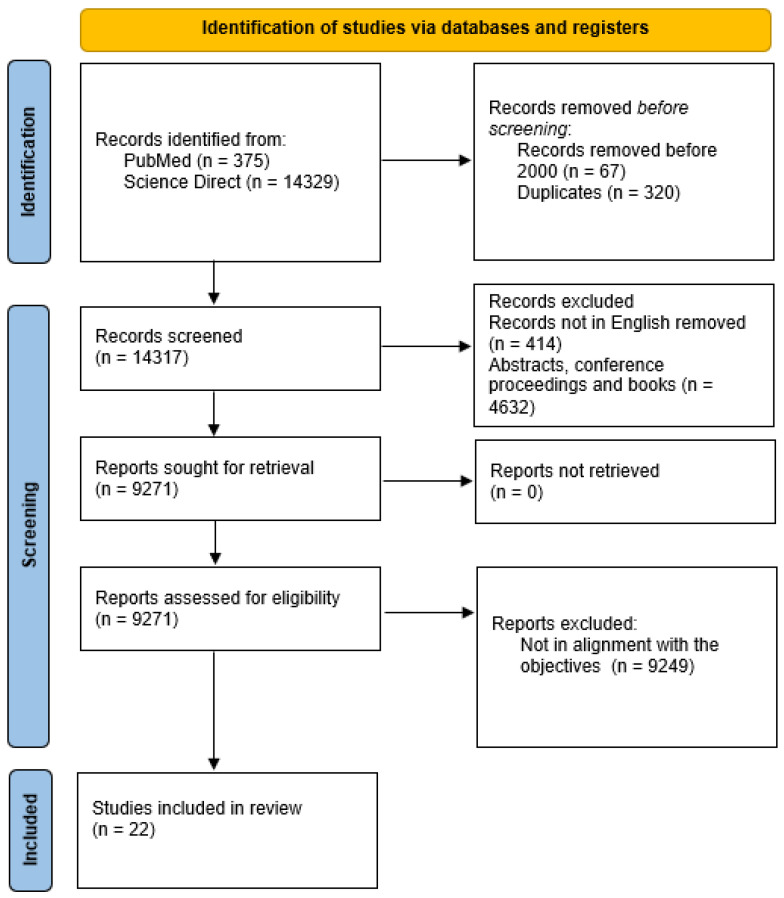
PRISMA flowchart.

**Figure 2 ijms-25-06430-f002:**
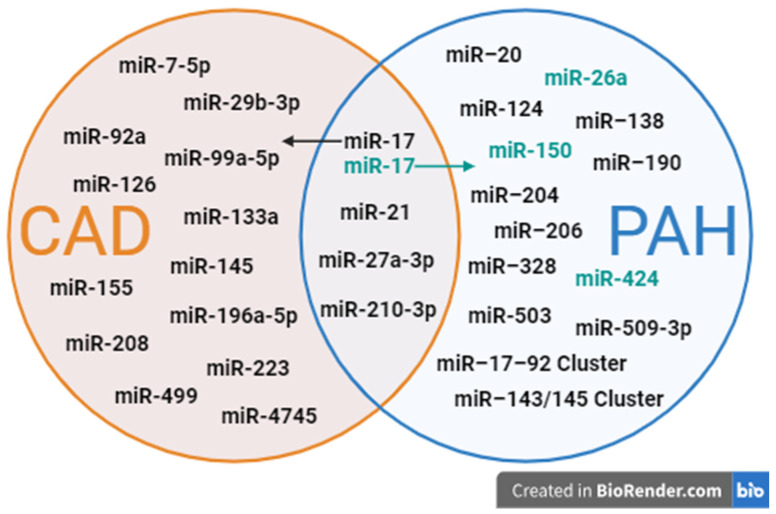
Comparative analysis of miRNAs in CAD and PAH: diagnostic and therapeutic potential. Note: CAD: coronary artery disease; PAH: pulmonary arterial hypertension; miR: MicroRNA; orange circle: mRNAs associated with CAD; blue circle: mRNAs associated with PAH; intersection (overlap): mRNAs shared between CAD and PAH, indicating potential roles in both conditions; black text: potential diagnostic value; green text: potential therapeutic value.

**Table 1 ijms-25-06430-t001:** The MMAT summarizes the included studies.

Category of Study Designs	Methodological Quality Criteria	Responses			
		Yes	No	Can’t Tell	Comments
**Screening questions** **(for all types)**	S1. Are there clear research questions?	✔			
	S2. Does the collected data allow me to address the research questions?	✔			
	Further appraisal may not be feasible or appropriate when the answer is ‘No’ or ‘Can’t tell’ to one or both screening questions.				
1. Qualitative	1.1. Is the qualitative approach appropriate to answer the research question?	✔			Shi et al., 2015 [9]; Santulli et al., 2016 [10]; Welten et al., 2016 [11]; Fang and Yeh, 2015 [12]; Henning, 2021 [13]; Ali et al., 2016 [14]; Buie et al., 2016 [15]; Sayed et al., 2014 [16]; Udali et al., 2013 [17]; Siasos et al., 2020 [18]; Johnson, 2019 [19]; Bienertova-Vasku et al., 2015 [20]
	1.2. Are the qualitative data collection methods adequate to address the research question?	✔			
	1.3. Are the findings adequately derived from the data?	✔			
	1.4. Is the interpretation of results sufficiently substantiated by data?	✔			
	1.5. Is there coherence between qualitative data sources, collection, analysis, and interpretation?	✔			
2. Quantitative randomized controlled trials	2.1. Is randomization appropriately performed?	✔			Patterson et al., 2020 [21]; Kanuri et al., 2018 [22]; Zhu et al., 2019 [23]; McCaffrey et al., 2023 [24]
	2.2. Are the groups comparable at baseline?	✔			
	2.3. Are there complete outcome data?	✔			
	2.4. Are outcome assessors blinded to the intervention provided?	✔			
	2.5. Did the participants adhere to the assigned intervention?	✔			
3. Quantitative non-randomized	3.1. Are the participants representative of the target population?	✔			Yao et al., 2018 [25]; Tomé-Carneiro et al., 2013 [26]; Li et al., 2019 [27]; Witarto et al., 2023 [28]; Tang, 2020 [29]; Agiannitopoulos et al., 2021 [30]
	3.2. Are measurements appropriate regarding both the outcome and intervention (or exposure)?	✔			
	3.3. Are there complete outcome data?	✔			
	3.4. Are the confounders accounted for in the design and analysis?	✔			
	3.5. During the study period, is the intervention administered (or exposure occurred) as intended?	✔			

**Table 2 ijms-25-06430-t002:** Characteristics of the included studies.

Nature of Study	Total Number of Patients	Intervention	Conclusions	References
Observational Study	98	Comparing miRNA profiles in PBMCs of CHD patients with and without HF and evaluate the significance of DEMs for HF risk in CHD patients.	CHD patients with and without HF could be differentiated according to PBMC miRNA profiles, and the combination of PBMC miR19b-5p, miR-221, miR-25-5p, and hypertension correlates with an increased HF risk in CHD patients.	Yao et al., 2018 [25]
Review Article	-	Summarizing the current knowledge regarding the mechanisms of miRNAs in cardiovascular remodeling, focusing specifically on hypertension.	The multiplexed detection of circulating miRNAs should be combined to compensate for the traits according to the study goal and specific miRNA features.	Shi et al., 2015 [9]
Observational Study	35	Examining molecular changes in PBMCs after one-year daily intake of 8 mg RES-enriched grape extract (GE-RES) in hypertensive male T2DM patients.	Long-term grape extract with RES reduces pro-inflammatory cytokine expression and inflammation-related miRs in immune cells of T2DM hypertensive patients, supporting an immunomodulatory treatment effect.	Tomé-Carneiro et al., 2013 [26]
Pilot Study	32	Assessing miRNA expression profiles in patients with or without CAD as selected by coronary CT angiography (CTA).	miRNA expression patterns in whole blood as selected on the basis of coronary CTA and risk scores vary significantly depending on the subject phenotype. Thus, profiling miRNA may improve early detection of CAD.	Patterson et al., 2020 [21]
Cohort study	437	Next-generation miRNA sequencing to identify CAD patients at risk of recurrent myocardial infarction.	MiRNA next-generation sequencing demonstrates altered the fingerprint profile of whole-blood miRNA expression among subjects with subsequent recurrent thrombotic events on standard medical therapy (‘non-responders’), as compared to subjects with no recurrent cardiovascular events.	Kanuri et al., 2018 [22]
Cohort study	65	High-throughput sequencing (NGS) was initially employed to compare circulating miRNA expression profiles in uLMCAD patients to that in patients without CAD to identify candidate miRNA biomarkers.	Circulating miR-182-5p and miR-5187-5p were suitable diagnostic biomarkers for uLMCAD, both potentially providing diagnostic information for discriminating uLMCAD patients from non-CAD population prior to invasive diagnostic coronary angiography (CAG).	Zhu et al., 2019 [23]
Review Article	-	Examining the complex interactions linking miRs, expression of genes, and molecular pathways leading to endothelial dysfunction.	The potential therapeutic use of miRs is currently being explored through several approaches, including inhibition and over-expression, in many cardiovascular disorders.	Santulli et al., 2016 [10]
Review Article	-	Discussing the multifactorial role of miRNAs and miRNA clusters that were reported to play a role in multiple forms of vascular remodeling and are clearly linked to cardiovascular disease.	Understanding the contribution of these miRNAs to the entire spectrum of vascular remodeling processes is important, especially as these miRNAs may have great potential as therapeutic targets for treatment of various cardiovascular diseases.	Welten et al., 2016 [11]
Review Paper	-	Examining the function and mechanisms of miRs, which are highly expressed in various cells types, especially endothelial and smooth muscle cells, which are closely involved in the process of vascular remodeling.	miRs play crucial roles in vascular remodeling.	Fang and Yeh, 2015 [12]
Review Paper	-	Examining the role of exosomes and miRNAs in normal myocardium, myocardial injury and infarction, atherosclerosis, and the importance of circulating miRNAs as biomarkers of cardiac disease.	Increased understanding of exosome content and function will facilitate their development as novel therapeutic strategies for the treatment of patients with atherosclerotic cardiovascular diseases.	Henning, 2021 [13]
Investigational study	24	Investigating the beneficial effect of miR-150 on PH in hypoxia-induced rats, PASMCs, and PAECs.	miR-150 protects against hypoxia-induced pulmonary vascular remodeling, fibrosis, and abnormal proliferation of PASMCs and PAECs, which suggests miR-150 as a promising therapeutic target for PH.	Li et al., 2019 [27]
Review Paper	-	Determining the number of miRNAs involved in pathophysiology of various cardiovascular diseases, and diagnostic and therapeutic potentials of miRNAs in these diseases.	Extensive research is required to explore the therapeutic and diagnostic values of miRNAs as successful as classical approaches.	Ali et al., 2016 [14]
Review Paper	-	Examining the role of miRNAs as either biomarkers or potential contributors to the pathophysiology of these aforementioned risk factors.	Patients with CAD have differentially expressed miRNAs due to a variety of signaling pathways activated during disease processes.	Buie et al., 2016 [15]
Review Paper	-	Examining more recent data regarding circulating miRNAs and their potential roles in diagnosis, prognosis, and therapeutic strategies for cardiovascular diseases.	Circulating miRNAs are resistant to endogenous ribonuclease activity and can be present in human plasma or serum in a remarkably stable form.	Sayed et al., 2014 [16]
Review Paper	-	Examining the role of epigenetics, from DNA methylation to miRNAs, in major cardiovascular diseases such as ischemic heart disease, hypertension, heart failure, and stroke.	DNA methylation as well as other epigenetic mechanisms, such as post-translational histone modifications and non-coding RNA-mediated mechanisms including miRNAs, are fascinating, novel approaches to be explored for a better understanding of cardiovascular disease pathogenesis and for the possible definition of useful, unique biomarkers for disease.	Udali et al., 2013 [17]
Cohort study	177	Employing Illumina RNAseq and network co-expression analysis to identify systematic changes underlying CAD.	The pattern of changes is consistent with stress-related changes in the maturation of T and Treg cells, possibly due to changes in the immune synapse.	McCaffrey et al., 2023 [24]
Bioinformatics study	2542	Investigating the diagnostic performance of circulating miRNAs in detecting subclinical carotid atherosclerosis.	Circulating miRNAs had excellent accuracy in detecting subclinical carotid atherosclerosis, suggesting their utilization as novel diagnostic tools.	Witarto et al., 2023 [28]
Diagnostic study	350	Evaluating the diagnostic value of circulating serum miR-509-3p in PAH (pulmonary arterial hypertension) with congenital heart disease.	The expression of miR-509-3p decreased in the serum of patients with PAH along with congenital heart disease.	Tang, 2020 [29]
Review Article	-	Presenting the most recent data concerning the role of miRNAs as potential novel biomarkers for cardiovascular disease.	The discovery of stable circulating miRNAs launches a new generation of the most promising biomarkers for CVD.	Siasos et al., 2020 [18]
Review Article	-	Knowledge on tissue and circulating miRNA expression changes identified to be involved in four key cardiovascular diseases; atherosclerosis, abdominal aortic aneurysms, restenosis, and pulmonary arterial hypertension.	Utilising a custom selection of miRNA inhibitors and/or mimics may provide an appealing therapeutic approach to manage single and co-existing cardiovascular diseases and their clinical complications.	Johnson, 2019 [19]
case-control study	400	Investigating whether four well-studied miRNA polymorphisms in non-Caucasian populations, namely miR146a G>C (rs2910164), miR149 C>T (rs2292832), miR196a2 C>T (rs11614913), and miR499 A>G (rs3746444), contribute to the risk for the development of premature CAD in the Greek population.	At least two of the studied polymorphisms, miR196a2 C>T (rs11614913) and miR499 A>G (rs3746444), as well as the miR146C–miR149C–miR196T–miR499G allele combination, could represent useful biomarkers of CAD and/or MI susceptibility in the Greek population.	Agiannitopoulos et al., 2021 [30]
Review Article	-	Examining the role of miRNAs in the development of PAH as well as their potential use as biomarkers and therapeutic tools in both experimental PAH models and in humans.	The current rapid development of our understanding of the actual biological roles of miRNAs in PAH development as well as partial successes in the field of RNAi therapy clearly justify an optimistic approach to future developments.	Bienertova-Vasku et al., 2015 [20]

## Data Availability

No new data were created or analyzed in this study. Data sharing is not applicable to this article.

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
