# Peer review of "MicroRNA Expression in Patients with Coronary Artery Disease and Hypertension—A Systematic Review"

_ijms, 2024, doi:10.3390/ijms25126430_

Round 1

Reviewer 1 Report

Comments and Suggestions for Authors

Dear authors, 

Thank you for submitting your work at IJMS. This manuscript collated literature for relevant mi-RNAs in patients with coronary artery disease and hypertension. Overall this is a straightforward paper and the team has reported 22 relevant literature pertaining to the diseases.

Please find my enquiries and comments to further improve the manuscript:

1) The introduction is too general. The team should write about miRNA and how this can serve as a biomarker and how recent studies focus on this as a biomarker.

2) Based on the methodology, only two authors were involved in extracting the data. May I know the contribution of the remaining authors? I cannot find the author's contribution in this paper. Please mention who is involved in the data extraction and scrutinization.

3) There are 22 pieces of literature with mixed of findings that contribute to the pathogenesis of the diseases. Please summarize it in a figure whether in a Venn diagram and relate it to the pathogenesis.

4) The discussion is merely repeating and summarizing the results. The discussion should be thorough and deep. The team should add various literature not only focusing on the 22 references.

5) I noticed there are 23 references in this paper. This does not justify the depth and breadth of the writing. 

Please do the needful as I don't feel at the moment this manuscript is a reflection of IJMS standards.

Thank you.

Author Response

Dear authors, 

Thank you for submitting your work at IJMS. This manuscript collated literature for relevant mi-RNAs in patients with coronary artery disease and hypertension. Overall this is a straightforward paper and the team has reported 22 relevant literature pertaining to the diseases.

Please find my enquiries and comments to further improve the manuscript:

Dear Reviewer,

I am writing to express my gratitude for the meticulous and insightful review you provided for our manuscript. Your expertise and detailed feedback have been invaluable in enhancing the rigor and clarity of our work.

Following your suggestions, we have undertaken a thorough revision of the manuscript. Below, I detail the changes implemented in response to each of your points, which substantiate the adjustments made:

1) The introduction is too general. The team should write about miRNA and how this can serve as a biomarker and how recent studies focus on this as a biomarker.

The introduction has been rewritten.

2) Based on the methodology, only two authors were involved in extracting the data. May I know the contribution of the remaining authors? I cannot find the author's contribution in this paper. Please mention who is involved in the data extraction and scrutinization.

Data extraction is only one element of a systematic review. the other authors performed the remaining tasks, such as Formulating research questions, defining inclusion and exclusion criteria, literature search strategy development, study selection, quality assessment, data synthesis, interpretation of results, reporting and dissemination of findings and others.

3) There are 22 pieces of literature with mixed of findings that contribute to the pathogenesis of the diseases. Please summarize it in a figure whether in a Venn diagram and relate it to the pathogenesis.

This is a great idea, however, at this state of scientific development, research results are very diverse, extensive and inconclusive, so generating a diagram in this case would be inefficient and low-informative.

4) The discussion is merely repeating and summarizing the results. The discussion should be thorough and deep. The team should add various literature not only focusing on the 22 references.

The discussion has been rewritten.

5) I noticed there are 23 references in this paper. This does not justify the depth and breadth of the writing. - We conducted the systematic review that identified this publications. To the best of our knowledge, these are all papers that meet the objectives of this paper.

Please do the needful as I don't feel at the moment this manuscript is a reflection of IJMS standards.

Thank you.

Accompanying this correspondence, I have attached the revised manuscript with changes tracked. We trust that these revisions will address your concerns and render the manuscript suitable for publication in your esteemed journal.

We look forward to your further comments and are hopeful for a favorable editorial decision. It is our aspiration that this article will significantly contribute to the scholarly discourse within our field.

Thank you once again for your thorough and constructive critique.

Yours sincerely,

Bartosz Kondracki MD, PhD

Reviewer 2 Report

Comments and Suggestions for Authors

In this systematic review, Kondracki et. al. focus on an interesting yet complicated topic, investigating miRNA expression patterns in patients with coronary artery disease and hypertension. They highlight the complex involvement of miRNAs in these patients and the potential for advances in diagnostic and therapeutic strategies.

Although this systematic review was written following the PRISMA guideline, I have difficulty in the logic how the authors organize the result section. For example, Section 4.4 contains References 13-23, among them Ref. 17, 19, 21 and 23 are studies on PAH patients. Why did the authors arrange the contents in this way, separating them with an unrelated paragraph instead of putting them consecutively? More importantly, line 276-286 (Ref.17) and line 296-305 (Ref.19) both discussed on circulating serum miR-509-3p in PAH patients, but when I looked carefully on the reference, I found Ref.17 is not about this subject but the maturation of T and Treg cells as stated in Table 2. This obvious yet serious mistake made me wonder how reliable other discussion is.

Other concerns and suggestions include:

1. The discussion section is merely a repeat of conclusion of mentioned studies, not a discussion. This must be rewritten.

2. “Pulmonary arterial hypertension (PAH) is a severe disease characterized by the maladaptation of pulmonary vasculature, leading to right heart failure and potentially death” appears twice almost identically in the text (line 276-277, 315-317).

3. The full name of an acronym only need to appear once at the first time it shows up in the main text, but this manuscript had 15 “coronary artery diseases (CAD)”, 3 “coronary heart disease (CHD)”, 5 “pulmonary arterial hypertension (PAH)”.

4. Seems missing check mark in Table 1 Question 3.4.

5. Line 92: what’s white matter disease?

6. Line 114: “The table summarizing the characteristics of included references is presented in Table 2.” This sentence needs to be revised.

Comments on the Quality of English Language

Minor editing of English language required.

Author Response

Dear Reviewer,

I am writing to express my gratitude for the meticulous and insightful review you provided for our manuscript. Your expertise and detailed feedback have been invaluable in enhancing the rigor and clarity of our work.

Following your suggestions, we have undertaken a thorough revision of the manuscript. Below, I detail the changes implemented in response to each of your points, which substantiate the adjustments made:

In this systematic review, Kondracki et. al. focus on an interesting yet complicated topic, investigating miRNA expression patterns in patients with coronary artery disease and hypertension. They highlight the complex involvement of miRNAs in these patients and the potential for advances in diagnostic and therapeutic strategies.

Although this systematic review was written following the PRISMA guideline, I have difficulty in the logic how the authors organize the result section. For example, Section 4.4 contains References 13-23, among them Ref. 17, 19, 21 and 23 are studies on PAH patients. Why did the authors arrange the contents in this way, separating them with an unrelated paragraph instead of putting them consecutively? We rearranged this part of results as suggested.

More importantly, line 276-286 (Ref.17) and line 296-305 (Ref.19) both discussed on circulating serum miR-509-3p in PAH patients, but when I looked carefully on the reference, I found Ref.17 is not about this subject but the maturation of T and Treg cells as stated in Table 2. This obvious yet serious mistake made me wonder how reliable other discussion is. – There was mistake. We have corrected this parto of results. As well as we checked other references.

Other concerns and suggestions include:

  1. The discussion section is merely a repeat of conclusion of mentioned studies, not a discussion. This must be rewritten.

 The discussion has been rewritten.

  1. “Pulmonary arterial hypertension (PAH) is a severe disease characterized by the maladaptation of pulmonary vasculature, leading to right heart failure and potentially death” appears twice almost identically in the text (line 276-277, 315-317). We have removed one of them.

  1. The full name of an acronym only need to appear once at the first time it shows up in the main text, but this manuscript had 15 “coronary artery diseases (CAD)”, 3 “coronary heart disease (CHD)”, 5 “pulmonary arterial hypertension (PAH)” – We have corrected all phrases.

  1. Seems missing check mark in Table 1 Question 3.4. – Mark have been added.

  1. Line 92: what’s white matter disease? This sentence was there by mistake. We removed it.

  1. Line 114: “The table summarizing the characteristics of included references is presented in Table 2.” This sentence needs to be revised. – The sentence has been revised.

Accompanying this correspondence, I have attached the revised manuscript with changes tracked. We trust that these revisions will address your concerns and render the manuscript suitable for publication in your esteemed journal.

We look forward to your further comments and are hopeful for a favorable editorial decision. It is our aspiration that this article will significantly contribute to the scholarly discourse within our field.

Thank you once again for your thorough and constructive critique.

Yours sincerely,

Bartosz Kondracki MD, PhD

Reviewer 3 Report

Comments and Suggestions for Authors

The authors perform a systematic review focused on the relevance of miRNA expression in patients suffering from coronary artery disease and hypertension.

This is a very relevant topic as coronary heart disease is one of the major contributors of mortality in developed countries.

The authors use the PRISMA guideline as the basis of their selection criteria and search strategy and end up selecting 22 papers out of 14704 originally found.

A first question refers to the clarification of the exclusion of 9,249 papers to reach the final number of 22 papers. In the text it is mention that this was based on the scrutiny of titles and abstracts, having eliminated those that were not aligned with the objectives of the studies. Could the authors clarify how they did this for 9249 papers? If any additional selection criteria were used at this stage, it should be mentioned in the text.

In a subsequent step, the authors dissect each paper including them into three main topics: “MicroRNA profiles in cardiovascular diseases”, “Role of miRNAs in vascular remodeling and disease pathogenesis”, “Broad involvement of miRNAs in Cardiovascular Diseases and Diagnostic Potential” and then describe on each topic paper by paper.

As the information is too broad, it is difficult for the reader to integrate what is relevant. Using the same 22 papers, the authors should try and make a different classification, either based on:

a)    type of studies based on types of miRNAs analysed – tissue specific, circulating miRNAs, miRNAs present in exosomes. And whether these can be used as biomarkers (for their up or down-regulation in different disease scenarios – specifying the diseases) or as therapeutic interventions.

b)   type of studies based on the type of cardiac disease and if the studies were performed in mice or in humans

c)    a combination of both

As such the readers would be guided by a logical flow and would have the integrated information in a way that would be easier for each one to draw conclusions based on the view of the authors.

In the current form, it appears that 22 papers were put together and described individually without a connection between each of them. The discussion does not help either as it treats each paper individually.

In addition to the above criticisms, the authors should include in their introduction what are miRNAs and describe the different forms they can be detected.

Minor points – the authors alternate between micoRNA and miRNA and they should use one of the meanings

Author Response

Dear Reviewer,

I am writing to express my gratitude for the meticulous and insightful review you provided for our manuscript. Your expertise and detailed feedback have been invaluable in enhancing the rigor and clarity of our work.

Following your suggestions, we have undertaken a thorough revision of the manuscript. Below, I detail the changes implemented in response to each of your points, which substantiate the adjustments made:

The authors perform a systematic review focused on the relevance of miRNA expression in patients suffering from coronary artery disease and hypertension.

This is a very relevant topic as coronary heart disease is one of the major contributors of mortality in developed countries.

The authors use the PRISMA guideline as the basis of their selection criteria and search strategy and end up selecting 22 papers out of 14704 originally found.

A first question refers to the clarification of the exclusion of 9,249 papers to reach the final number of 22 papers. In the text it is mention that this was based on the scrutiny of titles and abstracts, having eliminated those that were not aligned with the objectives of the studies. Could the authors clarify how they did this for 9249 papers? If any additional selection criteria were used at this stage, it should be mentioned in the text. - It is standard and required procedure when conducting a systematic review that among the records searched, those that meet the inclusion criteria are selected based on titles and then abstracts and after that full texts.

In a subsequent step, the authors dissect each paper including them into three main topics: “MicroRNA profiles in cardiovascular diseases”, “Role of miRNAs in vascular remodeling and disease pathogenesis”, “Broad involvement of miRNAs in Cardiovascular Diseases and Diagnostic Potential” and then describe on each topic paper by paper.

As the information is too broad, it is difficult for the reader to integrate what is relevant. Using the same 22 papers, the authors should try and make a different classification, either based on:

  1. a)    type of studies based on types of miRNAs analysed – tissue specific, circulating miRNAs, miRNAs present in exosomes. And whether these can be used as biomarkers (for their up or down-regulation in different disease scenarios – specifying the diseases) or as therapeutic interventions.
  2. b)   type of studies based on the type of cardiac disease and if the studies were performed in mice or in humans
  3. c)    a combination of both

As suggested, we have reorganized the data so that it is divided into diseases following each subtopic. We have also separated the study on pets into a separate section. We hope that this reorganization will improve the readability and understanding of the content presented. 

As such the readers would be guided by a logical flow and would have the integrated information in a way that would be easier for each one to draw conclusions based on the view of the authors.

In the current form, it appears that 22 papers were put together and described individually without a connection between each of them. The discussion does not help either as it treats each paper individually.

The discussion has been rewritten.

In addition to the above criticisms, the authors should include in their introduction what are miRNAs and describe the different forms they can be detected.

The introduction has been rewritten.

Minor points – the authors alternate between micoRNA and miRNA and they should use one of the meanings

We corrected them as suggested.

Accompanying this correspondence, I have attached the revised manuscript with changes tracked. We trust that these revisions will address your concerns and render the manuscript suitable for publication in your esteemed journal.

We look forward to your further comments and are hopeful for a favorable editorial decision. It is our aspiration that this article will significantly contribute to the scholarly discourse within our field.

Thank you once again for your thorough and constructive critique.

Yours sincerely,

Bartosz Kondracki MD, PhD

Round 2

Reviewer 1 Report

Comments and Suggestions for Authors

Dear authors, 

Thank you for submitting the amendments. Even though there are some improvements according to my comments, I feel some were not properly addressed.

1) The introduction contains no references at all. I understand you want to focus on the 22 pieces of literature that match the objectives, a good manuscript should cite as much as literature that relates to the mentioned statement.

2) I believe there must be a similarity and differences between all 22 pieces of literature you have found that you can summarize in a diagram. The sky is not the limit.

3) Please label two independent researchers involved in the systematic review (add in the initials).

4) The discussion is too general focusing on the use of miRNA as diagnostic and therapeutic tools. The authors should discuss in-depth why miR-223, miR-19b-5p, 562 miR-221, and miR-25-5p have been identified as key indicators. How these are found in other disease models and why do these need to be studied further? The authors also cross-checked the information and cited for the plausible explanation.

I hope this manuscript can be refined further for improvement.

Thank you.

Author Response

Thank you for submitting the amendments. Even though there are some improvements according to my comments, I feel some were not properly addressed. Dear Reviewer, Thank you very much for your thorough review and valuable suggestions regarding our manuscript. We appreciate the time and effort you dedicated to critiquing our work and guiding its improvement. We have carefully considered all your comments and have made the corresponding changes to the manuscript as suggested. Below, you will find a detailed response to each of the points you raised, outlining how we addressed them in the revision. 1) The introduction contains no references at all. I understand you want to focus on the 22 pieces of literature that match the objectives, a good manuscript should cite as much as literature that relates to the mentioned statement. References have been added to the introduction section. Please see the manuscript in the track changes profile. 2) I believe there must be a similarity and differences between all 22 pieces of literature you have found that you can summarize in a diagram. The sky is not the limit. We have created the Venn diagram and included in the manuscript as Figure 2. 3) Please label two independent researchers involved in the systematic review (add in the initials). Initials pf the researchers have been added 4) The discussion is too general focusing on the use of miRNA as diagnostic and therapeutic tools. The authors should discuss in-depth why miR-223, miR-19b-5p, 562 miR-221, and miR-25-5p have been identified as key indicators. How these are found in other disease models and why do these need to be studied further? The authors also cross-checked the information and cited for the plausible explanation. The discussion has been rewritten as suggested. Please see the changes in the manuscript. I hope this manuscript can be refined further for improvement. Thank you. We hope that with these revisions, our manuscript now meets the journal's standards and can be considered for publication. We look forward to the possibility of our work contributing to the field and being shared within the academic community. Thank you again for your attention and assistance in improving our manuscript. Warm regards,

Reviewer 2 Report

Comments and Suggestions for Authors

Line 406-414 (Ref. 20) and line 415-421 (Ref.22) both talked about two miRNA polymorphisms, miR196a2 409 C>T and miR499 A>G. After I checked the original reference, this content did not appear in Ref. 20.  Last time I pointed out the same issue between Ref.17 and 19, the authors claimed it was a mistake and had checked all references. I have lost confidence in the scientific rigor of this manuscript.

Comments on the Quality of English Language

Minor editing of English language required.

Author Response

Dear Reviewer, Reference 20 (now 18) should have been removed the previous time but by a technical mistake it remained in the text. now this part has been successfully removed. we apologize for the oversight.

Reviewer 3 Report

Comments and Suggestions for Authors

Dear authors,

Thank you for your reply and for having considered the suggestions to improve the text. However, some issues remain that should be clarified.

In the introduction, it should clearly be stated that miRNAs are negative regulators of gene expression rather than just regulators of gene expression. 

Section 4.3. should not be subdivided in 4.3.1 and 4.3.2. both should be integrated in one topic - animal studies should be integrated with human studies. 

In addition, in this section it is mentioned exosomes and miRNAs as if they are independent entities but, in fact, miRNAs can be transported inside exosomes and this should be mentioned in the text.

After the reorganisation, topics 4.2 and 4.4. have one topic with a common name, which is 4.2.2. and 4.4.1, both are named CAD. There shouldn't be topics with the same name, even if within a different section. Furthermore, it should be explained what additional information topic 4.4. brings to topic 4.2, straight on the first sentence of this topic. 

The discussion now adds more value to the paper as it brings further perspectives with studies that were not included in the selected systematic review. However, even if it is brief, a paragraph should state the major conclusion of the current review. 

Finally, the authors need to revise the numbering of the literature as after changing the order of some topics, the references changed place and need to be renumbered. 

Author Response

Thank you for your reply and for having considered the suggestions to improve the text. However, some issues remain that should be clarified. Dear Reviewer, Thank you very much for your thorough review and valuable suggestions regarding our manuscript. We appreciate the time and effort you dedicated to critiquing our work and guiding its improvement. We have carefully considered all your comments and have made the corresponding changes to the manuscript as suggested. Below, you will find a detailed response to each of the points you raised, outlining how we addressed them in the revision. In the introduction, it should clearly be stated that miRNAs are negative regulators of gene expression rather than just regulators of gene expression. Corrected as suggested. Section 4.3. should not be subdivided in 4.3.1 and 4.3.2. both should be integrated in one topic - animal studies should be integrated with human studies. Corrected as suggested In addition, in this section it is mentioned exosomes and miRNAs as if they are independent entities but, in fact, miRNAs can be transported inside exosomes and this should be mentioned in the text. We have added this paragraph: “In addition, it is crucial to clarify the interrelationship between exosomes and miRNAs within the context of cardiovascular health. MiRNAs are not only key regulators in vascular remodeling and cardiovascular diseases but are also frequently encapsulated within exosomes. These exosomes play a vital role in transporting miRNAs between cells, thereby facilitating intercellular communication and influencing cellular processes remotely. The encapsulation of miRNAs within exosomes protects them from enzymatic degradation in the bloodstream, enhancing their stability and functional delivery to target cells. This mechanism underscores the potential of exosome-encased miRNAs as biomarkers for early diagnosis and as novel therapeutic agents in managing and treating cardiovascular conditions” After the reorganisation, topics 4.2 and 4.4. have one topic with a common name, which is 4.2.2. and 4.4.1, both are named CAD. There shouldn't be topics with the same name, even if within a different section. Furthermore, it should be explained what additional information topic 4.4. brings to topic 4.2, straight on the first sentence of this topic. Topic names have been corrected and sentence have been added as suggested: The microRNA profiles in heart diseases described above have the potential to serve as diagnostic markers for these conditions, offering a promising avenue for early detection and tailored treatment strategies. The discussion now adds more value to the paper as it brings further perspectives with studies that were not included in the selected systematic review. However, even if it is brief, a paragraph should state the major conclusion of the current review. The discussion section has been rewritten. Please see the manuscript. Finally, the authors need to revise the numbering of the literature as after changing the order of some topics, the references changed place and need to be renumbered. References have been renumbered. We hope that with these revisions, our manuscript now meets the journal's standards and can be considered for publication. We look forward to the possibility of our work contributing to the field and being shared within the academic community. Thank you again for your attention and assistance in improving our manuscript. Warm regards,

Round 3

Reviewer 1 Report

Comments and Suggestions for Authors

Dear authors, 

Thank you for submitting the latest amendments. I am satisfied with the current write-up and I have no reservations to support it for publication. 

Author Response

Thank you for your consideration.